# Biomimetic Oil-in-Water Nanoemulsions as a Suitable Drug Delivery System to Target Inflamed Endothelial Cells

**DOI:** 10.3390/nano14151286

**Published:** 2024-07-31

**Authors:** Elena Lagreca, Elisabetta Caiazzo, Concetta Di Natale, Valentina Mollo, Raffaele Vecchione, Armando Ialenti, Paolo Antonio Netti

**Affiliations:** 1Center for Advanced Biomaterials for Health Care (CABHC), Istituto Italiano di Tecnologia, Largo Barsanti e Matteucci 53, 80125 Naples, Italy; elena.lagreca@iit.it (E.L.); valentina.mollo@iit.it (V.M.); paolo.netti@iit.it (P.A.N.); 2Dipartimento di Chimica dei Materiali e Produzioni Industriali (DICMaPI), University of Naples Federico II, P. le Tecchio 80, 80125 Naples, Italy; 3Department of Pharmacy, School of Medicine and Surgery, University of Naples Federico II, 80131 Naples, Italy; elisabetta.caiazzo@unina.it (E.C.); armando.ialenti@unina.it (A.I.); 4School of Infection & Immunity, College of Medical, Veterinary and Life Sciences, University of Glasgow, Glasgow G12 8TA, UK; 5Interdisciplinary Research Centre on Biomaterials (CRIB), University of Naples Federico II, P. le Tecchio 80, 80125 Naples, Italy

**Keywords:** biomimetic nanocarrier, surface functionalisation, nano-emulsions

## Abstract

Currently, the biomimetic approach of drawing inspiration from nature has frequently been employed in designing drug nanocarriers (NCs) of actively target various diseases, ranging from cancer to neuronal and inflammation pathologies. The cell-membrane coating can confer upon the inner nanomaterials a biological identity and the functions exhibited by the cells from which the membrane is derived. Monocyte- and macrophage-membrane-coated nanomaterials have emerged as an ideal delivery system to target inflamed vasculature. Herein, we developed two biomimetic NCs using a human-derived leukaemia monocytic cell line (THP-1), either undifferentiated or differentiated by phorbol 12-myristate 13-acetate (PMA) into adherent macrophage-like cells as membrane sources for NC coating. We employed a secondary oil-in-water nano-emulsion (SNE) as the inner core, which served as an optimal NC for high payloads of lipophilic compounds. Two different biomimetic systems were produced, combining the biomimetic features of biological membranes with the physicochemical and nano-sized characteristics of SNEs. These systems were named Monocyte NEsoSome (M-NEsoSome) and Macrophage NEsoSome (M0-NEsoSome). Their uptake ability was investigated in tumour necrosis factor alfa (TNFα)-treated human umbilical vein endothelial cells (HUVECs), selected as a model of inflamed endothelial cells. The M0 membrane coating demonstrated accelerated internalisation compared with the monocyte coating and notably surpassed the uptake rate of bare NCs. In conclusion, M0-NEsoSome NCs could be a therapeutic system for targeting inflamed endothelial cells and potentially delivering anti-inflammatory drugs in vascular inflammation.

## 1. Introduction

The application of nanotechnology offers significant opportunities for diagnosing and treating a broad spectrum of diseases, including inflammatory ones [1,2]. The vascular endothelium has long been recognised for its crucial role in regulating inflammation, acting as a critical interface between circulating immune cells and inflamed tissues [3]. In this context, targeting the inflamed vascular endothelium may offer a promising avenue for effective therapeutic interventions in diseases such as atherosclerosis and autoimmune disorders [4]. Nanocarriers (NCs) have shown a passive targeting affinity for endothelial cells, which experience a notable increase in permeability during both acute and chronic inflammation [5]. Nevertheless, the passive targeting effect of NCs may still not be sufficient to meet the effective and safe therapeutic standards of heterogeneous therapy.

Recently, there has been growing interest in utilising cell-membrane-coated materials as an innovative approach to target inflamed vascular endothelium [6].

Bioinspired approaches utilise engineered biological materials for drug delivery, effectively bypassing the immune system and overcoming biological barriers. Cell-membrane-coated nanocarriers (CM-NCs) exhibit vital features such as biocompatibility, extended circulation times, immune evasion, and enhanced drug targeting and accumulation. These nanocarriers can be customised for specific applications, including tracking and diagnostics, using various cell types like red and white blood cells [7], cancer cells [8], platelets [9], bacteria [10], and stem cells [11]. The choice of cell type or membrane is critical for ensuring targeted delivery and minimising adverse interactions within the body [12]. The cell membrane layer is highly versatile and can be modified for specific purposes, such as being labelled for tracking and diagnostics [11,13,14]. These kinds of NCs demonstrate a vast and diversified field of applications, including chemotherapy, neurological disease, immune therapy, and gene therapy.

These biomimetic nanomaterials combine the inherent characteristics of membranes from specific cell types, such as leukocytes, with the adaptability of synthetic NCs, providing a unique platform for the precise and efficient delivery of therapeutic agents to selectively target inflamed endothelial cells.

Monocytes and macrophages are key players in the immune system, being involved from the initiation to the resolution of inflammation [15]. Recently, nanoparticles coated with their extracted membranes have gained attention as a delivery system for targeting the inflamed vasculature. These systems can directly adhere to inflammatory endothelial cells, blocking the essential monocyte entry pathway into the injury site. Macrophage-membrane-coated nanoparticles can enhance the accumulation of nanoparticles at specific inflammatory sites by protecting them from immune cell phagocytosis and clearance [16]. Currently, monocyte and macrophage-membrane-coated nanoparticles are in the embryonic phase of development, presenting significant potential for clinical application [17].

One remarkable application is in the field of tumour treatment. For instance, macrophages have been successfully employed for cancer targeting, particularly in treating glioblastoma multiforme (GBM), a highly immunosuppressive tumour. Ju et al. developed A2-MPC, a drug delivery system that combines macrophage membranes with the angiopep-2 peptide. Genetically engineered macrophages with specific surface markers loaded with nanoparticles represent a breakthrough in tumour therapy. Wang et al. introduced PD-1-MM@PLGA/RAPA, which combines macrophage membranes expressing PD-1 with PLGA nanoparticles loaded with rapamycin. This combination enhances bioavailability and bioactivity, improving tumour targeting and treatment effectiveness. Similarly, Zhang et al. developed aPD-1-scFv NVs, macrophage-derived vesicles displaying anti-PD-1 antibodies and incorporating CPI-444 to counteract immunosuppressive adenosine. This combination significantly increased T-cell density and functionality, effectively suppressing tumour progression and metastasis [18].

This work aimed to develop oil-in-water secondary nano-emulsions (O/W SNEs) with a monocyte- or macrophage-membrane layer capable of retaining the properties of the cell source, with the potential to target inflamed endothelial cells in vitro.

O/W SNEs are a well-established nano-delivery system for lipophilic substances [19,20,21,22,23,24]. SNEs represent raw materials that are easily tunable for different applications, including oral and parenteral delivery systems. SNEs can be the starting building block for multilayer nano-emulsions [25,26], with properties related to the selected coating strategies (i.e., stimuli-responsive system [27,28], active targeting [29], blood-brain barrier (BBB) overcoming [30]). We refer to this type of NC with the term NEsoSome to highlight its nano-sized features joined with its lipophilic core, enabling it to carry both internal and external cargo. Previously, we proposed biomimetic NCs with a cancer-cell-membrane layer (CM-NEsoSome), which showed good stability and potential to be a suitable NC for tumour targeting [31]. Here, we focused on developing biomimetic nano-emulsions to reach endothelial cells, which act as a critical interface between circulating immune cells and inflamed tissues. We assembled two biomimetic NCs employing monocyte- and macrophage-membranes as external layers to achieve this. We selected THP-1 as a human monocyte cell line and used it to obtain human macrophage membranes. Specifically, THP-1 cells were differentiated into the macrophage-like phenotype (THP-1 macrophages) by incubation with phorbol 12-myristate-13-acetate (PMA) to express cell membrane markers and exploit macrophage “*homing*”. The two systems were assembled, employing THP-1 and PMA-differentiated THP-1 as membrane sources for the monocyte (M) and unpolarised macrophage (M0) NEsoSomes, respectively, indicated as M-NEsoSome and M0-NEsoSome. The systems were morphologically analysed by dynamic light scattering (DLS), ζ-potential, and cryo-electron transmission microscopy (cryo-TEM). The uptake ability of the systems was evaluated in a human umbilical vein endothelial cell (HUVEC) layer.

## 2. Material and Methods

### 2.1. Materials

Surfactant Lipoid E80 (egg lecithin powder 80–85% enriched with phosphatidylcholine (PC) and 7–9.5% content of phosphatidyl ethanolamine (PE)) was purchased from Lipoid GmbH (Ludwigshafen, Germany). Millipore Milli-Q water was used to prepare all nano-emulsions and solutions. Soybean oil (density at 20 °C of 0.922 g mL^−1^), fluorescein-isothiocyanate (FITC), and chitosan (Ct, LMW 90–150 kDa, DDA 84% determined via ^1^H-NMR) were purchased from Sigma Aldrich (Milan, Italy). Phosphate-buffered saline (PBS), glutaraldehyde, sodium cacodylate, osmium tetroxide, potassium ferrocyanide, Spurr’s resin, uranyl acetate, and copper grids were purchased from electron microscopy sciences (Societa Italiana Chimici, Rome, Italy).

Ethanol and tannic acid were purchased from Merck (Valsamoggia, Italy).

RPMI-1640 Glutamax (Gibco, Grand Island, NY, USA), fetal bovine serum (FBS, Gibco), antibiotics (100 U/mL penicillin, 100 mg mL^−1^ streptomycin, Gibco), phorbol 12-myristate 13-acetate (PMA) Hoechst 33342, trihydrochloride, trihydrate (Hoechst), and CellMask™ Orange plasma membrane stain (CellMask™-543) were purchased from Thermo Fisher Scientific and Life Technologies (Monza, Italy). VascuLife^®^ VEGF endothelial medium complete kit was purchased from CellSystems GmbH (Troisdorf, Germany). Sodium pyruvate, 2-mercaptoethanol, and gelatine were purchased from Merck (Milan, Italy). Recombinant human tumour necrosis factor alfa (TNFα) was purchased from Peprotech; RabMab recombinant anti-CD-55 antibody [EPR22362-255] (ab243231) 0.539 mg/mL and recombinant anti-CD-33 antibody mouse monoclonal [7G9] 0.937 mg/mL were purchased from Abcam (Prodotti Gianni, Milan, Italy).

### 2.2. Methods

#### 2.2.1. THP-1 Culture and Activation to Unpolarised Macrophages

THP-1 cells at passage 3 were seeded at a density of 5 × 10^5^ cells/mL in RPMI-1640 Glutamax supplemented with 10% inactivated FBS, 1% antibiotics, 1 mM sodium pyruvate, and 0.05 mM 2-mercaptoethanol. The cells were sub-cultured in a T-75 cell culture flask for membrane extraction under a humidified atmosphere containing 5% CO_2_ at 37 °C. The medium was refreshed every 2–3 days. For differentiation of unpolarised macrophages, namely M0, and membrane extraction, cells were seeded in a 6-well plate and treated with 50 ng/mL PMA for 16 h, followed by 48 h resting to obtain M0 [32] (Figure 1). Cell differentiation after PMA treatment was verified by evaluating cell adhesion and spreading under an optical microscope. More in detail, cell differentiation was estimated by Invitrogen™ EVOS™ Digital Colour Fluorescence Microscope (Thermo Fisher Scientific and Life Technologies, Waltham, MA, USA), immunofluorescence analysis (see Section 2.2.2 on immunofluorescence (IF)), and transmission electron microscope (TEM) analysis for both suspension cells (PMA-free THP-1, M) and adherent cells (PMA-differentiated THP-1, M0). Before extracting the plasma membrane, the PMA-differentiated and undifferentiated THP-1 cells were treated with 1× PBS solution containing 0.01% Hoechst and 0.1% CellMask™-543. This solution was added to each flask/plate and left for 10 min to stain the nuclei and plasma membrane, followed by three washings with 1× PBS.

#### 2.2.2. Immunofluorescence (IF)

Immunofluorescence (IF) was used to confirm the differentiation of THP-1 monocytes into M0 macrophages. Macrophages were cultured in WillCo-Dish glass-bottom dishes (WillCo Wells B.V., Amsterdam, The Netherlands) with a height of 0.17 mm and a diameter of 1.2 mm, following established protocols. To prepare the monocytes (PMA-free THP-1, M), media changes and washing steps were performed by centrifugation at 1500 rpm for 5 min. The final cell pellet was adhered to the glass bottom dish (WillCo Wells B.V., Amsterdam, The Netherlands) by drying a drop of PBS containing 100,000 cells. The cells underwent three washes with 1× PBS, then fixation with 4% paraformaldehyde (PAF) for 10 min, and another three with 1× PBS. Before immunostaining, cells were treated with 100 mM glycine for 5 min and incubated with 1% bovine serum albumin (BSA) for 30 min at room temperature (RT). Each incubation step was followed by three 5-min washes with 1× PBS. The cells were then incubated overnight at 4 °C with primary antibodies, Rabbit Monoclonal Recombinant Anti-CD-55 and Anti-CD-33 antibody [4B4F12], diluted 1:100 in 1% BSA. Afterwards, the cells were washed three times with 1× PBS and incubated with a secondary antibody solution diluted 1:300 in 1% BSA for 1 h at RT in the dark, followed by 3 washes with 1× PBS. For counterstaining, the cells were treated with a 1:1000 dilution of Hoechst stain for 10 min and washed three times with 1× PBS. Control samples included those without primary antibodies, only counterstains, and dye-labelled secondary antibodies. Confocal dishes were air-dried and rehydrated for the monocyte samples with a mounting medium of 80% glycerol and PBS. Monocytes and macrophages were visualised using a Confocal Zeiss Axio Observer Z1 microscope (Carl Zeiss, Goettingen, Germany) with a 63× oil objective at 1024 × 1024 pixels resolution. Image analysis was conducted using ImageJ^®^ software 1.54f, Wayne Rasband contributors National Institute of Health, USA.

#### 2.2.3. Transmission Electron Microscopy (TEM Analysis)

Monocytes (PMA-free THP-1, M) were centrifuged at 1200 rpm for 5 min during each washing and media-changing step. All steps were initially conducted in a Petri dish for macrophage (differentiated THP-1, M0) sample preparation. Samples were removed, centrifuged at 1200 rpm for 5 min, and dehydrated. Samples were fixed in 2.5% glutaraldehyde in 0.1 M sodium cacodylate buffer at 4 °C overnight, followed by three washes in the same buffer (5 min for each step). Post-fixation was performed in a solution of 1% osmium tetroxide/1% potassium ferrocyanide in sodium cacodylate for 1 h at 4 °C. After three washes in sodium cacodylate, specimens were stained overnight at 4 °C in the dark with a 1% uranyl acetate aqueous solution. Following staining, samples were washed in chilled water, incubated with 0.15% tannic acid aqueous solution for 3 min, and then rinsed again in chilled distilled water before dehydration. Dehydration was performed using an ascending ethanol series (30%, 50%, 70%, 95% (2 times), and 100% (3 times)), with each step lasting 10 min on ice. The final ethanol step was performed at room temperature before overnight infiltration in a 1:1 mixture of Spurr’s resin and ethanol. Samples were embedded in fresh Spurr’s resin for 2 days prior to polymerisation at 70 °C. Each sample was sectioned with a diamond knife (Diatome) using an ultramicrotome FC7-UC5 Leica (Leica Biosystems Nussloch GmbH. Nussloch, Germany) to a thickness of 80 nm, and sections were collected on 200-mesh-thin bar copper grids. Imaging was performed using Cryo-TEM Tecnai G2, 20, 200 KV-FEI Thermofisher Company (Eugene, ON, USA) equipped with a CCD Camera 2 HS, Eagle and Veleta side-view camera, Olympus; Cryoholder, Gatan, at 120 KV, with a magnification range of 5 KX and 50 KX.

#### 2.2.4. HUVEC Cultures

Human umbilical vein endothelial cells (HUVECs), passage 4, were grown with VascuLife^®^ VEGF Endothelial Medium Complete Kit in a T-75 cell culture flask pre-treated for 30 min with 1% (*w*/*v*) gelatine, in a humidified controlled atmosphere with 5% of CO_2_, at 37 °C. The medium was changed every 2–3 days. HUVECs were cultured in 8-well plates and stimulated with TNFα (10 ng/mL) for 24 h. The 8 wells were pre-treated with 1% (*w*/*v*) gelatine for 30 min.

#### 2.2.5. Membrane Extraction and Characterisation

Plasma membranes were obtained from PMA-differentiated THP-1 cells (macrophages) and, as a control, from PMA-free THP-1 cells (monocytes) following the procedure reported by Profeta et al. with some modifications [31]. After cell staining, monocyte cells were washed with PBS and collected by centrifugation at 1200 rpm for 5 min, then suspended in hypotonic lysing buffer at a 1:10 ratio of pellet to lysis buffer. Macrophage-adherent cells were detached using a cell scraper, washed with PBS, and collected by centrifugation at 1200 rpm for 5 min, then suspended at a 1:10 ratio of pellet to hypotonic lysis buffer. The hypotonic lysis buffer consisted of 20 mM Tris-HCl (pH 7.5), 10 mM KCl, 2 mM MgCl_2_, and 20 mM sucrose. Both cell types were disrupted by thorough pipetting and spinning the solution at 3200× *g* for 5 min. The pellet was re-dissolved in hypotonic lysis buffer, pipetted, and centrifuged again at 3200× *g* for 6 min. The supernatants were collected, mixed, and centrifuged at 20,000× *g* for 20 min at 4 °C. Membrane pellets from macrophages (M) and monocytes (M0) were obtained by final centrifugation at 100,000× *g* for 15 min at 4 °C. The pellet was then redispersed in 1 mL of 1× PBS, characterised, and used as purified M and M0 membranes for subsequent experiments.

We conducted the membrane extraction protocol in triplicate to ensure the reproducibility and purity of the final samples. Each step of the purification process was analysed using a confocal microscope (Leica Microsystems TCS SP5 II, Wetzlar, Germany) with a 25 × 0.8 N water immersion objective. Images were captured at a resolution of 1024 × 1024 pixels. For protein quantification of the cell membranes, we utilised the Bicinchoninic Acid (BCA) assay (Merck, Milan, Italy), following the manufacturer’s instructions [33,34]. We measured absorbance at 562 nm with an EnSpire^®^ Multimode Plate Reader (PerkinElmer, Inc. Waltham, MA, USA), and the titration curve is presented in Appendix A. To evaluate protein-membrane integrity, we used circular dichroism (CD). The CD spectra of the membrane solutions (3 µg/mL and 2.4 µg/mL for monocytes and macrophages, respectively) were recorded using a Jasco J-1500 spectro-polarimeter (J-1500-150, Tokyo, Japan) in a 1.0 cm path-length quartz cell. We recorded CD spectra at 25 °C in the far UV region from 260 to 200 nm, averaging over three scans and correcting for blanks. The spectra deconvolution was performed using the BeStSel program (BeStSel™ (2014–2024)–ELTE Eötvös Loránd University, Budapest, Hungary) [35], as reported previously [36,37].

#### 2.2.6. SNE Preparation: Chitosan-Layered NEs (Ct-NEs)

The inner core was a stabilised nano-emulsion (SNE), specifically an oil-in-water (O/W) nano-emulsion (NE), referred to as the “primary nano-emulsion”, coated with a polyelectrolyte of opposite charge, chitosan (Ct). This nano-complex is designated as Ct-NE.

To prepare the oil phase, 5.8 g of Lipoid E 80 was dissolved in 24 mL of soybean oil at 60 °C using an immersion sonicator (Ultrasonic Processor VCX500 Sonic and Materials, Hielscher Ultrasonics, Teltow, Germany). The oil phase was then added dropwise to the water phase (Milli-Q water) and sonicated again to form a pre-emulsion. This process was performed at low temperatures using an ice bath. The pre-emulsion was then subjected to high-pressure homogenisation at 2000 bar using a Microfluidizer^®^ (Microfluidics^TM^, M110PS, Westwood, MA, USA) to reduce the initial droplet size significantly [38].

The polyelectrolyte coating was achieved by depositing fluorescently labelled chitosan (Ct) onto the O/W NEs following a previously developed method [30]. Specifically, Ct was chemically labelled with fluorescein 5(6)-isothiocyanate (FITC) as described in earlier work [19]. To prepare the Ct-FITC, 100 mg of Ct (0.50 mmol) was dispersed in 10 mL of 0.1 M acetic acid solution. After dissolution, a FITC solution (5.0 mg in 500 μL of dimethyl sulfoxide, DMSO) was added dropwise. The reaction was allowed to proceed overnight at room temperature; then, the product was precipitated by adjusting the pH to 10 with NaOH and washed several times with water via centrifugation (Thermo-Scientific SL16R, Thermo Fisher Scientific, Waltham, MA, USA) at 9000 rpm for 15 min. The purified product was freeze-dried (Freeze Dryer CHRIST Alpha 1–4 LSC, Martin Christ Gefriertrocknungsanlagen GmbH, Osterode am Harz, Germany) for 48 h, yielding Ct-FITC, which was used to assemble the SNEs. To assemble the Ct-NEs, a 0.1 M acetic acid solution of Ct-FITC (0.2 *w*/*v*%, pH 4) was rapidly added to the O/W NEs (20 wt% oil) under vigorous stirring and maintained for 15 min to ensure uniform Ct deposition. The Ct-NEs were then re-dispersed using a high-pressure homogeniser at 700 bar for approximately 100 continuous cycles and re-processed under the same conditions after one week. The final product was stored at room temperature. The concentrations of oil and Ct in the final Ct-NEs were 1 *w*/*v*% and 0.01 *w*/*v*%, respectively, with a final pH of 4.24.

#### 2.2.7. Biomimetic NEsoSome Assembly and Characterisation

The isolated macrophage or monocyte membranes were deposited onto the surface of the Ct-NEs, exploiting the electrostatic interaction between the positively charged Ct-NEs and the negatively charged cell membranes. Initially, the cell membranes were re-dispersed by passing them through an Avanti Polar Lipids extruder (Merk, Milan, Italy) equipped with a 0.4 µm membrane (PC membranes 0.4 µm, Avanti Polar Lipids, Merk, Milan, Italy) 21 times. These extruded membranes were then deposited onto the surface of the Ct-NEs, following an adapted CM-NEsoSome procedure [31]. Briefly, Ct-NEs were quickly added, under vigorous stirring, to the extruded cell membrane solution and kept under stirring for 15 min to allow uniform cell-membrane deposition, and then co-extruded for 21 passes with an Avanti Polar Lipids extruder using a 0.4 µm membrane (PC membranes 0.4 µm, Avanti Polar Lipids). Biomimetic NEsoSomes with monocyte and macrophage membranes were both characterised by DLS and cryo-TEM according to the procedure reported in previous work [39].

Ct-NEs, the M-NEsoSome, and M0-NEsoSome were characterised by measuring their size, polydispersity index (PdI), and ζ-potential using a dynamic light scattering (DLS) instrument (Zetasizer ZS, Nanoseries ZEN 3600, Malvern Instruments Ltd., Malvern, UK, λ = 632.8 nm). All samples were diluted to a droplet concentration of approximately 0.01 wt% using Milli-Q water. Measurements were taken at a detecting angle of 173° with a default refractive index ratio of 1.5900. Three runs of 100 s each were performed per sample for particle size distribution. ζ-potential analysis was conducted with 30 runs for each measurement. Cryo-TEM samples of M-NEsoSome and M0-NEsoSome were prepared using the plunge-freezing technique with the Vitrobot Mark IV (FEI Company, Eugene, ON, USA). For each sample, 3 μL was dispensed onto a 200-mesh Quantifoil copper grid in the Vitrobot chamber. The volume was reduced by blotting for 1 s with filter paper to create a final thin film approximately 100–200 nm thick. To prevent sample evaporation, the Vitrobot was set to 95% humidity and 4 °C with a waiting time of 60 s before plunging into liquid propane. After grid transfer into liquid nitrogen, each sample was mounted on a Gatan Cryo holder and observed using a transmission electron microscope (TEM) TECNAI G2-20 (FEI Company, Eugene, ON, USA) equipped with a Gatan CCD camera 2HS, in Cryo mode. Imaging was performed in low-dose mode at 200 kV with magnifications ranging from 20,000 to 50,000.

#### 2.2.8. In Vitro Accumulation Analysis in HUVECs

The targeting ability of the prepared nano-complexes (NCs) to activated endothelial cells (ECs) was assessed by simulating the inflammatory activation of ECs using TNFα-stimulated human umbilical vein endothelial cells (HUVECs) [40,41]. As described above, 5 × 10^4^ HUVECs were seeded on 8-well plates and inflamed with 10 ng/mL TNFα for 24 h. Cells were incubated with M0-NEsoSomes, M-NEsoSomes, Ct-NEs, and cell medium alone as a positive control (Appendix A), at a final concentration of 0.2 wt% of oil in water concentration for 30 min, 24 and 48 h at 37 °C in 5% CO_2_. To prove the selectivity for inflamed endothelial cells, uptake experiments were also made in not-inflamed HUVECs. Then, samples were washed three times with 1× PBS to remove the non-internalized compounds and fixed with 4% PAF for 10 min. Finally, cells were incubated with Hoechst diluted 1:10,000 in 1× PBS for 10 min RT for cell nuclei staining. Samples were observed by Confocal Zeiss Axio Observer Z1 microscope (Carl Zeiss, Goettingen, Germany) using a 20× objective at a resolution of 1024 × 1024 pixels. All experiments were performed in triplicate.

## 3. Result and Discussion

### 3.1. THP-1 Activation and Cell Membrane Extraction

The THP-1 cell line was chosen as the membrane source for both monocytes and, once activated, the macrophages. The THP-1 cells were differentiated by 16 h of PMA treatment to induce the first step of macrophage maturation/differentiation, resulting in unpolarised macrophages (M0), as outlined in Figure 1a–c.

The differentiation was evaluated through cell morphological changes via microscope evaluation, TEM analysis, and immunofluorescence. After 16 h of PMA treatment, as expected, THP-1 cells exhibited the typical hallmarks of macrophage-like cells, including cell adhesion, spread morphology, increased granularity, increased cytoplasm, more cytoplasmic organelles (such as mitochondria and lysosomes), and an irregular nucleus shape [42], as detected by optical and TEM images (Figure 2b,d). Specifically, macrophage-like cells (Figure 2b,d) can be typified by a wide cytoplasm abundant in lysosomes and mitochondria. Frequently, the macrophage nucleus is larger, polymorphic, and multilobate (Figure 2d) [43,44]. THP-1 differentiation was also assessed with immunofluorescence analysis, evaluating CD-55 and CD-33 expression on the cell membrane of both THP-1 and PMA-differentiated THP-1 cells. Confocal images of THP-1 monocytes and PMA-treated THP-1 M0 macrophage-like cells were captured, showing the CD-55 signal in green, the CD-33 in red, and the nuclei in blue. Untreated THP-1 monocytes displayed a red signal corresponding to CD-33 but not the CD-55 green signal (Appendix A). In contrast, CD-55 was expressed in PMA-treated THP-1 cells, while the CD-33 signal intensity decreased (Appendix A). These findings align with previous studies and demonstrate a consistent differentiation process of monocytes into M0 macrophages [45].

Cell membranes were isolated from both differentiated THP-1 (Macrophage, M0) and THP-1 (Monocyte, M) cells according to the procedure reported by Profeta et al. [31], with some modifications (Figure 1a–c). This technique involves a gentle rupture of hypotonic-treated cells. Cell contents were removed and isolated cell membrane “ghosts” through repetitive sequential washing and centrifugation at varying speeds. The pellets and the supernatants from each step of the purification process were collected and examined by confocal microscopy. Before membrane extraction, cell nuclei (blue) and plasma cell membranes (red) were stained (Figure 3). Figure 3a,c show the first supernatant containing isolated cell nuclei for both cell sources (Figure 3a, M; Figure 3c, M0). Figure 3b,d show the final pellet of the monocyte and macrophage membrane extractions. The absence of the blue nuclear signal in both samples indicates a reasonable degree of purification for both cell lines.

After the final purification step, the protein content of the membranes from both cell sources was evaluated using the BCA essay. The results revealed 3 µg/mL concentrations for monocytes and 3.4 µg/mL for macrophages. To further validate the BCA and confocal analysis and assess whether the membrane proteins preserved their secondary structure, CD analysis was performed [46]. The recorded CD spectra, shown in Figure 4, confirm that the extraction and purification processes were performed correctly for both cell membrane sources, as no DNA bands typically found at 260 nm were detected. Moreover, the membrane proteins retained their secondary structure, exhibiting characteristic mixed α-helix, β-sheet, and random conformations with minima at 222 and 205 nm and a maximum of 195 nm. These results were further supported by the deconvolution analysis detailed in Appendix A.

### 3.2. Biomimetic NEsoSome Preparation and Characterisation

Once the cell membranes were obtained, they were used to coat Ct-NEs, providing highly biomimetic NCs with active targeting capability. Ct-NEs (or NEsoSomes), which form the inner core of the final NCs, were prepared using a well-established procedure. The dimensions of Ct-NEs were 102.5 nm ± 0.96 with a polydispersity index (PdI) of 0.09 ± 0.01 (Figure 5, green line) and a ζ-potential of +30 mV. The membrane was quickly added to the Ct-NEs’ solution under vigorous stirring and kept under stirring for 15 min, followed by one extrusion cycle of 21 passages through a 100 nm polycarbonate filter to ensure uniform membrane coating. The biological membrane coating on Ct-NEs was driven by electrostatic and hydrophobic interactions between the negatively charged membrane and the positively charged Ct-NEs, as observed with other types of cell-membrane-coated materials [31,41]. The fate and behaviour of the NCs are closely related to particle size and ζ-potential values, which were therefore monitored throughout the process. All formulations exhibited a size distribution of less than 140 nm. In particular, DLS analysis revealed an increase in size of approximately 20 nm for both types of cell source (M0-NEsoSome: size 121.2 nm ± 1.2, PdI 0.14 ± 0.02, Figure 5, red line; M-NEsoSome: size 122 nm ± 1.23, PdI 0.15 ± 0.04, Figure 5, blue line) compared to Ct-NEs (size 102.5 nm ± 0.96, PdI 0.09 ± 0.01), consistent with the thickness of the macrophage-membrane coating [40]. Compared to Ct-NEs, the ζ-potential decreased by approximately 10 mV due to the typical negative charge of the phospholipid membrane of cell sources (Ct NEs +25 mV, M0-NEsoSome +11 mV and M-NEsoSome +12 mV, see the Appendix A). The overall positive charge is beneficial for cellular uptake, given that cell membranes are negatively charged [39]. Cryo-TEM analysis further validated the presence of the membrane coating (Figure 6), showing that both biomimetic NCs exhibited nanometric size and an external layer provided by the cell-membrane coating. The colloidal stability of both M and M0 formulations was evaluated, revealing consistent sizes over one month when stored at 4 °C, indicating excellent stability. These results suggest that both M-NEsoSome and M0-NEsoSome have similar physicochemical properties to uncoated NEsoSome (data reported in the Appendix A).

### 3.3. In Vitro Targeting of Endothelial Cells

The ability to target inflamed endothelial cells was evaluated by comparing the uptake of M0-NEsoSome, M-NEsoSome, and uncoated NEsoSome (Ct-NEs) at 0.2 wt% final oil-in-water concentration. Cell medium alone served as the positive control (Appendix A). This evaluation was conducted in both TNFα-treated HUVECs and control HUVECs at various time points (30 min, 24 h, and 48 h) using confocal microscopy. For tracking, the extracted cell membrane and Ct polymer were stained with CellMask™-543 (red) and FITC (green), respectively. Confocal microscopy was used to visualise their signals while interacting with cells. Therefore, Ct-NE corresponds to the green signal, whereas M- and M0-NEsoSomes correspond to the yellow hotspots given by merging the two fluorophores in the same pixel. Figure 7 illustrates the in vitro accumulation of NCs in TNFα-treated HUVECs at different time points. The results show that M0-NEsoSome uptake was detected as early as 30 min (Figure 7c, yellow spots, red box inset) and increased over time, as depicted in Figure 7f.

In detail, the images after 30 min of treatment highlighted a higher accumulation and uptake of M0-NEsoSome (Figure 7c, red box inset) with inflamed HUVECs compared to M-NEsoSome (Figure 7b, red box inset) and even more so compared to uncoated NEsoSome (Figure 7a, red box inset), which showed no accumulation and therefore internalisation at this time (the Appendix A). However, the yellow signal of the M-NEsoSome was evident after 24 h and 48 h, as highlighted in the red box insets in each figure (Figure 7e,h), suggesting a slower accumulation compared to M0-NEsoSome (Figure 7f,i). Monocyte-coated NCs in HUVECs were also investigated by Jiang et al., who observed that the uptake efficiency of the Monocyte Cell-Membrane-Coated 1,8-Cineole Biomimetic Delivery System (MM-CIN-BDS or BDS) surpassed that of uncoated NPs in LPS-activated HUVECs, despite uncoated NPs showing higher uptake efficiency in normal HUVECs. This difference is attributed to the expression of adhesion molecules (VCAM-1, ICAM-1, L-selectin, etc.) on LPS-activated HUVECs. These molecules facilitate the adhesion and binding of MM-CIN-BDS through antibody-antigen interactions, leading to increased internalisation of MM-CIN-BDS [47]. More in-depth, M-NEsoSome accumulation in stimulated HUVECs is intermediate between M0-NEsoSomes and Ct-NEs (Figure 7a,d,g; the Appendix A). These results can be explained by the behaviour of blood-circulating monocytes during inflammatory responses, as they are driven to the sites of inflammation where they differentiate into tissue macrophages. Following 24 h and 48 h of treatment, Ct-NEs (green spots, Figure 7d,g) could also internalise. The positive ζ-potential value of Ct-NEs could justify these results due to the high-charge-density areas on the cell surface, which can mediate the endocytosis of positively charged particles [48]. After 24 h and 48 h, both M0-(Figure 7f,i) and M-NEsoSomes (Figure 7e,h) showed an intense signal (yellow spots), although still stronger in the M0-NEsoSome. In the Appendix A, we report the plot of the mean fluorescence intensity normalised for the cell number. As we can see, after 24 h, it reached the plateau in terms of MFI and internalisation for both M0- and M-NEsoSomes (the Appendix A). Of course, in dynamic in vivo conditions, timing is crucial since a delay in NC internalisation may result in a washout of the NC itself, nullifying its effect.

In unstimulated HUVECs, the three types of NCs behaved similarly, with almost 5–20% internalisation in the first 30 min of treatment and intensification in the following 24 h (Figure 8f). Nonetheless, after 24 h and 48 h, all formulations were internalised in unstimulated HUVECs (Figure 8g–i; the Appendix A). This result demonstrates the macrophages’ greater affinity for inflamed endothelium than healthy endothelium. In unstimulated HUVECs, the uncoated nanocarrier also had an uptake capacity comparable to the membrane-coated analogues.

This different behaviour could be explained by the expression of fewer adhesion molecules in normal endothelial cells than in inflamed ones. Therefore, monocyte- and macrophage-coated NEsoSomes adhere poorly to them, resulting in slower accumulation and uptake than in the inflamed ECs model. These results highlighted the selectivity of the inflamed ECs for M0- and M-membrane-coated NCs compared to uncoated Ct-NEs. This effect is much more evident in the case of M0-membrane-coated NCs, as expected.

## 4. Discussion

Using cell-membrane-coated materials for targeting inflamed vascular endothelium offers several advantages over conventional drug delivery methods. Firstly, the biomimetic nature of these materials enhances biocompatibility and reduces immunogenicity, minimising adverse immune responses. Secondly, the surface proteins and receptors inherent in the cell membrane facilitate targeting specificity, enabling selective binding and uptake by inflamed endothelial cells while reducing off-target effects. It should be noted that an effective drug delivery system must be able to evade the immune system to ensure prolonged blood circulation and subsequent accumulation in the targeted tissue.

Thanks to their innate properties, researchers proposed the assembly of a leukocyte-membrane-derived NC, which proved capable of evading the immune system, crossing the biological barriers of the body, and localising at target tissues [41].

Macrophages are pivotal in innate immunity and are well-known for their phagocytic activity, antigen presentation, and adaptable phenotypes. Macrophage-membrane-coated nanoparticles are particularly effective in targeting chronic inflammatory sites such as cancer, gout, and atherosclerosis, neutralising inflammatory cytokines more efficiently than RBC-membrane-coated nanoparticles. Inflammation, a protective immune response, is implicated in numerous diseases and driven by pro-inflammatory cytokines like IL-1β, TNFα, and IL-6. Immune cell membranes from neutrophils and macrophages are increasingly employed in inflammation-targeted therapies [49]. Macrophage-membrane-coated nanocarriers excel in neutralising cytokines and precisely targeting inflammation sites compared to RBC-coated MOFs. Additionally, coating nanocarriers with macrophage membranes shows promise for other applications, such as cancer and sepsis treatment. For instance, research by Chen et al. revealed that tumour-associated macrophage (TAM) membranes from primary tumours possess unique antigen-homing capabilities and immune compatibility. TAM-membrane-coated nanoparticles can scavenge CSF1, thereby disrupting TAM–cancer cell interactions and enhancing the effectiveness of cancer immunotherapy [50]. Thamphiwatana et al. developed macrophage biomimetic nanoparticles by enveloping polymeric cores with macrophage-derived cell membranes for sepsis management. These nanoparticles mimic macrophages, binding and neutralising endotoxins that would otherwise trigger immune activation, thereby acting as macrophage decoys [51].

Overall, these studies demonstrate that cell-membrane cloaking of nanoparticles enhances targeted drug delivery for inflammation therapy. Here, we proposed the utilisation of two distinct cell-membrane-coated O/W secondary nano-emulsions, employing monocytes and macrophages from THP-1, a human monocytic cell line, as a cell membrane source to avoid the critical issues related to non-human cell-membrane sources. More specifically, we evaluated the assembly of the biomimetic nanocarrier employing different cell sources (i.e., monocytes and macrophages from THP-1). Using physical characteristics (DLS and cryo-TEM), we investigated and compared how the cell-membrane coating affected the final nanocarrier morphologies. As a result, we observed an increase in size values of around 20 nm due to cell-membrane layers, as evidenced through DLS and confirmed by Cryo-TEM. Cryo-TEM clearly showed the presence of cell-membrane layers. These results agree with our previous work [31]. The colloidal stability of both M and M0 formulations was investigated. Their sizes remained consistent over four weeks, indicating excellent stability when stored at 4 °C. These findings suggest that both M-NEsoSome and M0-NEsoSome possess similar physicochemical properties to uncoated NEsoSome. Notably, the sizes of these formulations remained stable and comparable to uncoated NEsoSome for up to 24 days before starting to increase (the Appendix A). We assessed the in vitro behaviour of the overall systems in both inflamed (TNFα-treated HUVECs) and healthy ECs layers (HUVECs). Our previous research [34] utilised similar nanocarriers with various cellular membranes. These studies and the present study included detailed uptake analyses, and we did not observe signs of cytotoxicity or mortality in normal cells. This provides a preliminary indication of the biocompatibility of our NCs. We plan to integrate dedicated cytotoxicity studies into our research framework together with efficacy texts to offer critical insights into their therapeutic potential and dosage optimisation. Results indicated that the macrophage membrane facilitated higher and faster accumulation in TNFα-treated HUVECs compared to M-NEsoSome and the membrane-uncoated NEsoSome (Ct-NEs). After 24 h, the MFI reached a plateau, indicating maximum internalisation for both M0- and M-NEsoSomes. The uptake efficiency of particles in the HUVECs and TNFα-treated HUVECs increased for both M0- and M-NEsoSomes, supporting the hypothesis that all formulations were fully internalised after 24 h. The highest intensity was observed for M0-NEsoSome in the TNFα-treated HUVECs. In healthy HUVECs, both biomimetic formulations (M0- and M-NEsoSomes) behaved similarly during the initial 30 min and up to 4 h; however, after 24 h, a slight difference was observed, with the M-NEsoSome showing better internalisation (data reported in the Appendix A).

It is well-known that HUVECs activated with TNFα overexpress VCAM-1, ensuring interaction specificity with integrin α4β1 on the macrophages [52]. Furthermore, selective accumulation of cell-membrane-coated materials in the TNFα-treated HUVEC layers was observed compared to non-TNFα-treated HUVEC layers. These results align with the natural behaviour of macrophages and monocytes, which can reach activated ECs due to the presence of cell adhesion molecules, particularly in the case of macrophages. Specifically, M0 macrophages represent the quiescent state of monocyte-derived macrophages (MDMs). Macrophage polarisation involves the adjustment of M0 macrophages to their surrounding environment, resulting in changes in phenotypes and diverse functions. Following acute injury, monocyte-derived macrophages, including M0 macrophages, migrate to the affected tissue and become predominant. M0 macrophages, being non-activated, are attracted to inflamed tissues in response to various signals. Our results are in line with previously reported data demonstrating that monocyte-membrane-coated rapamycin nanoparticles effectively penetrate inflamed endothelium [53]. In addition, macrophage-membrane-camouflaging of rapamycin or colchicine nanoparticles effectively protects nanoparticles from phagocytosis by macrophages and targets activated endothelial cells in vitro [54]. Moreover, these macrophage-coated nanoparticles showed efficient targeting and accumulation in atherosclerotic lesions in apolipoprotein-E knockout (ApoE^−/−^) mice fed a high-fat diet [54]. The therapeutic efficacy of macrophage-biomimetic drug delivery systems in atherosclerotic mouse models has also been confirmed in other studies [55,56].

Our results are in line with previously reported data demonstrating that monocyte-[53]. In addition, macrophage-membrane-camouflaging of rapamycin or colchicine nanoparticles effectively protects nanoparticles from phagocytosis by macrophages and targets in vitro [54]. Moreover, these macrophage-coated nanoparticles showed efficient targeting and accumulation in atherosclerotic lesions in apolipoprotein-E knockout (ApoE^−/−^) mice fed a high-fat diet [40]. The therapeutic efficacy of macrophage-biomimetic drug delivery systems in atherosclerotic mouse models has also been confirmed in other studies [55,56].

However, all findings related to monocyte- and macrophage-camouflaged nanoparticles targeting activated endothelial cells were derived from murine cell lines, which may not accurately represent human monocytes/macrophages. Utilising this system in non-human models could prove ineffective and may lead to issues such as immune responses or the lack of active targeting receptors for the nano-vectors.

To the best of our knowledge, this is the first study using human-macrophage-camouflaging nanoparticles to target in vitro endothelial cells. Our M0/M systems could serve as a suitable drug delivery system for lipophilic molecules due to the high payload capability of the inner oil core. Additionally, vegetable oil can be easily absorbed without the safety risks typically associated with solid nanocarriers. Notably, oil-in-water nano-emulsions are still stable enough to reach the target tissue in a stable form compared to other soft nanocarriers. However, this work represents a preliminary study, and these findings should be validated through in vivo analysis and immune evasion testing. Future investigations should also explore the utilisation of polarised macrophage membranes, such as M1 and M2 macrophages, or even an engineered membrane.

## 5. Conclusions

Nanotechnology presents a promising avenue for diagnosing and treating inflammatory diseases, focusing on targeting inflamed vascular endothelium for effective therapeutic interventions. NCs have shown potential in passively targeting endothelial cells experiencing increased permeability during inflammation, but their efficacy may not meet heterogeneous therapy standards. The recent interest in cell-membrane-coated materials offers a unique approach, combining the specific characteristics of cell membranes with the adaptability of synthetic NCs for precise drug delivery. Monocytes and macrophages, pivotal in the immune system, have gained attention for their membrane-coated nanoparticles’ potential in targeting inflamed vasculature, blocking monocyte entry, and enhancing nanoparticle accumulation at inflammatory sites. Herein, the development of O/W SNEs with monocyte or macrophage-membrane layers aims to target inflamed endothelial cells while utilising a well-established delivery system for lipophilic substances. These biomimetic nano-emulsions, termed NEsoSomes, were assembled using the human monocyte cell line THP-1 and differentiated like macrophages. They were morphologically analysed and evaluated for uptake in HUVEC layers, showing promising potential for targeted drug delivery in vitro.

## Figures and Tables

**Figure 1 nanomaterials-14-01286-f001:**
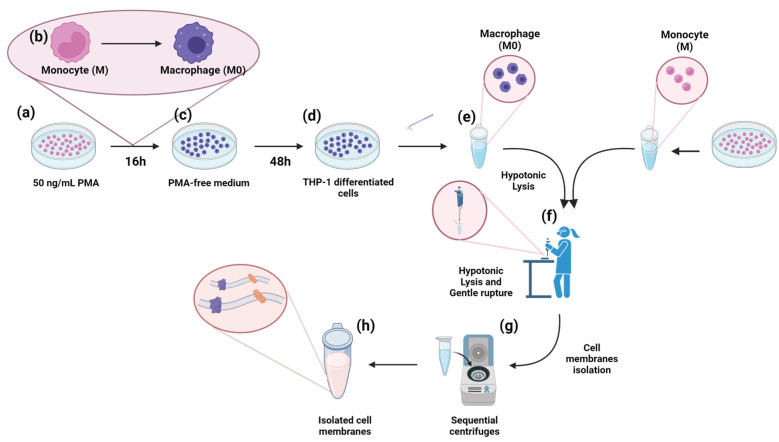
Schematic representation of THP-1 activation with PMA (**a**–**d**) and plasma membrane extraction procedure (**e**–**h**). Created with BioRendrer^®^.

**Figure 2 nanomaterials-14-01286-f002:**
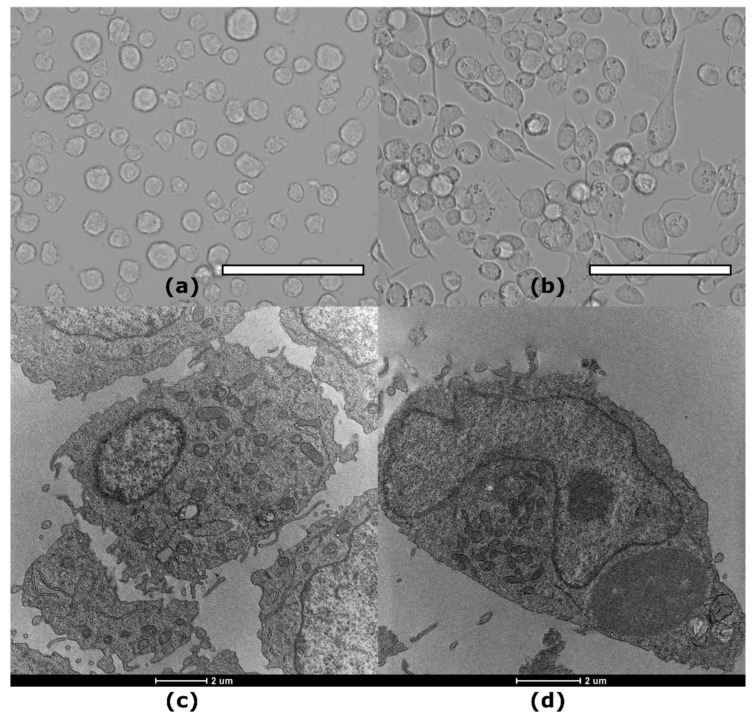
Morphological analysis of control and PMA-treated THP-1: top optical microscopy images of (**a**) THP-1 monocyte cells, (**b**) THP-1 PMA-activated macrophages for 16 h, scale bar 100 µm; below, TEM images of (**c**) THP-1 monocyte and (**d**) THP-1 PMA-activated macrophage for 16 h.

**Figure 3 nanomaterials-14-01286-f003:**
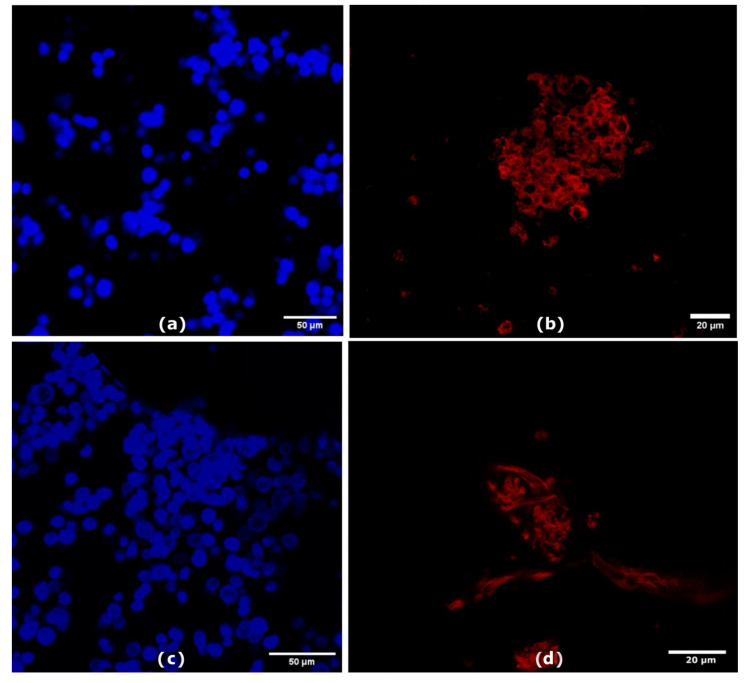
Confocal images of initial pellet with nuclei of (**a**) monocyte (M) and (**c**) macrophage (M0) scale bar 50 µm, and final membrane pellet of (**b**) monocyte (M) and (**d**) macrophage (M0) scale bar 20 µm.

**Figure 4 nanomaterials-14-01286-f004:**
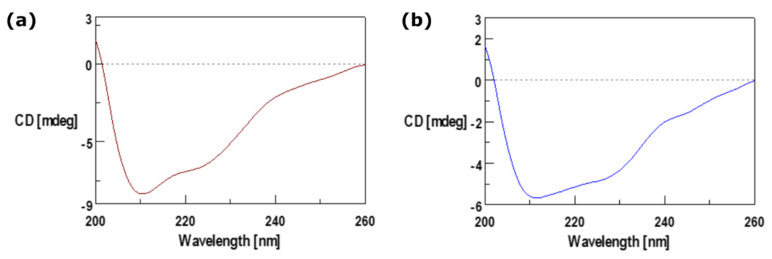
CD spectra for isolated plasma membrane of (**a**) monocytes and (**b**) macrophages.

**Figure 5 nanomaterials-14-01286-f005:**
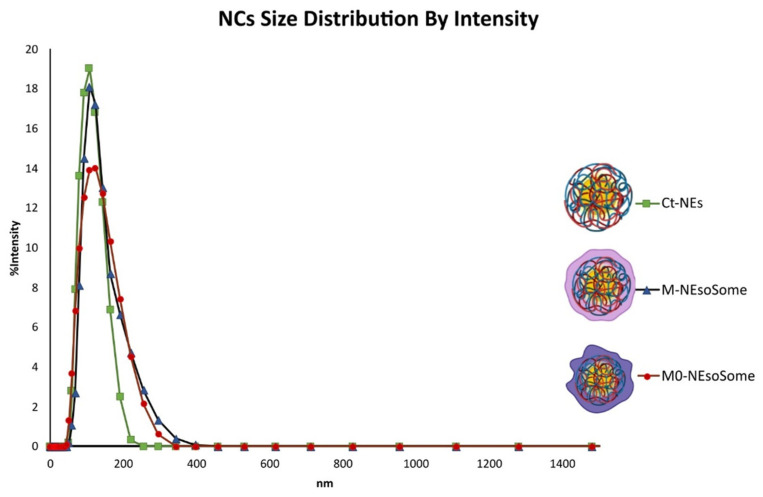
NCs size distribution by the intensity of Ct-NEs (green line and box), M-NEsoSome (blue line and triangle), and M0-NEsoSome (red line and circle).

**Figure 6 nanomaterials-14-01286-f006:**
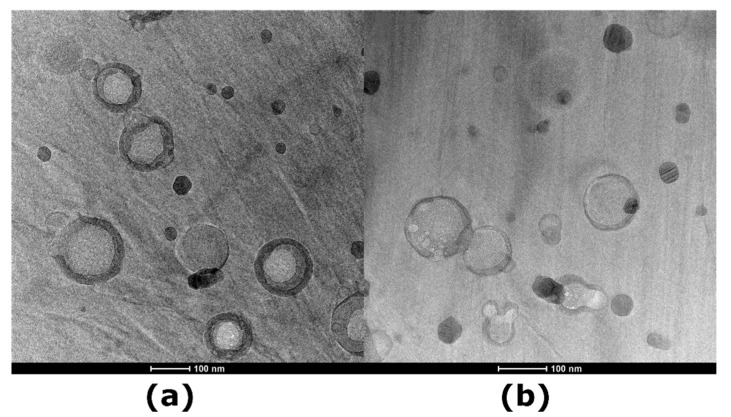
Cryo-TEM images of (**a**) M0 NEsoSome and (**b**) M-NEsoSome; scale bar 100 nm.

**Figure 7 nanomaterials-14-01286-f007:**
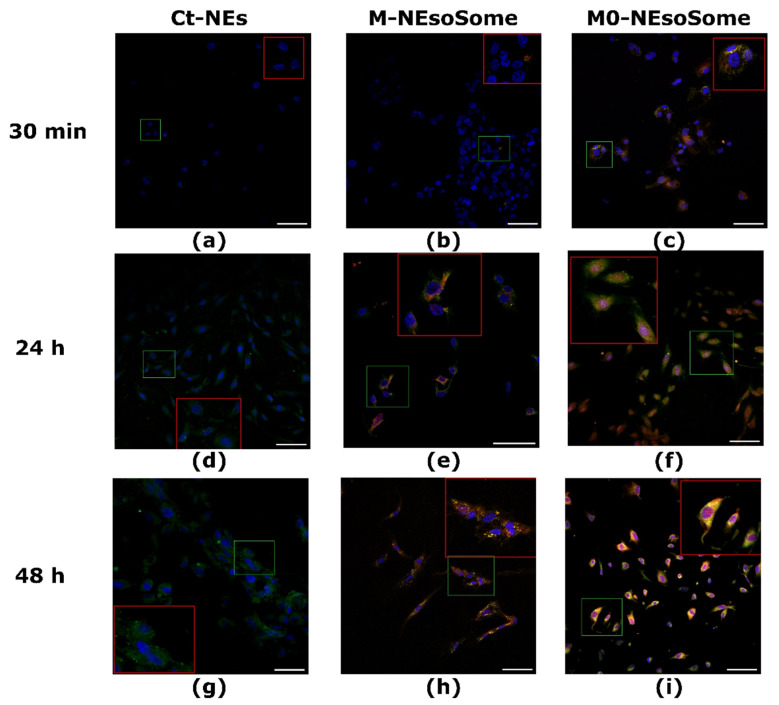
In vitro uptake of Ct-NEs (**a**,**d**,**g**), M-NEsoSome (**b**,**e**,**h**), and M0-NEsoSome (**c**,**f**,**i**) in TNFα-treated HUVECs.

**Figure 8 nanomaterials-14-01286-f008:**
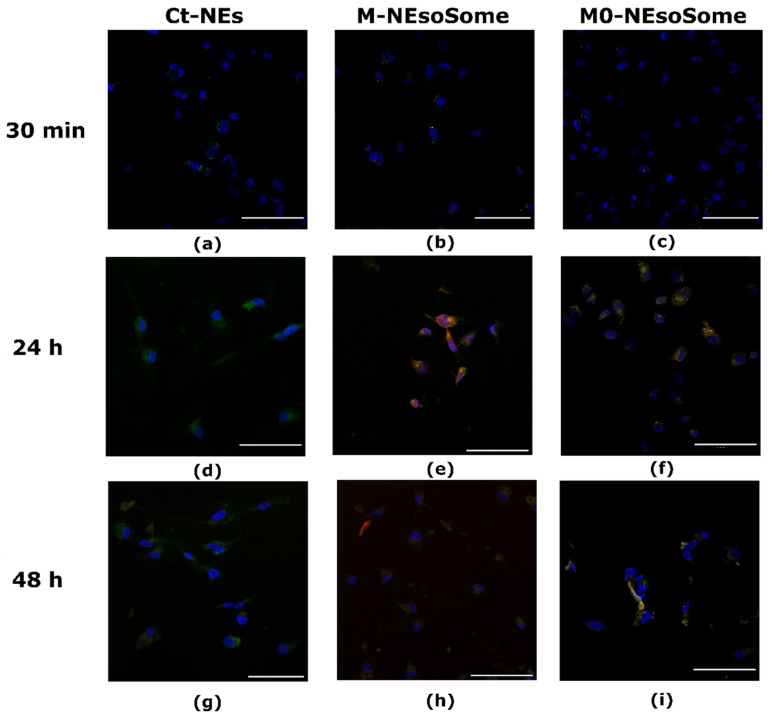
In vitro uptake of Ct-NEs (**a**,**d**,**g**), M-NEsoSome (**b**,**e**,**h**), and M0-NEsoSome (**c**,**f**,**i**) in HUVECs.

## Data Availability

The data presented in this study is contained within this article.

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
