# Peer review of "Biomimetic Oil-in-Water Nanoemulsions as a Suitable Drug Delivery System to Target Inflamed Endothelial Cells"

_nanomaterials, 2024, doi:10.3390/nano14151286_

Round 1

Reviewer 1 Report

Comments and Suggestions for Authors

This manuscript developed two biomimetic NCs using a human-derived leukaemia monocytic cell line (THP-1), either undifferentiated or differentiated by phorbol 12-myristate 13-acetate (PMA) into adherent macro-phage-like cells as membrane sources for NC coating. Then a secondary oil in water nanoemulsion (SNE) served as the inner core, which serves as an optimal NC for high payloads of lipophilic compounds. M0-NEsoSome NCs could be a therapeutic system for targeting inflamed endothelial cells and for the potential delivery of anti-inflammatory drugs in vascular inflammation. This work is herein recommended for consideration in Nanomaterials. Some issues should be addressed before acceptance.

1.      Cell membrane modification technology is the focus of drug design in this article. Therefore, it is recommended that the authors provide a comprehensive overview of cell membrane technologies in the Introduction section. This should include the current treatment status of cancer (Science Advances, 2024, 10, eadm9561; Small, 2024, 2404741), inflammation (Nature Communications, 2024, 15, 1042; Advanced Science, 2024, 2310211.), and other diseases (Biomaterials, 2024, 306, 122478; Exploration, 2024, 20230164; Advanced Science, 2023, 10, 2302131) using cell membrane-wrapped nano drugs administered via intravenous injection, oral ingestion, and other delivery routes. Additionally, they should summarize the advancement of current research methods in this field.

2.      The morphology and composition characterization of nano drugs need further refinement. For example, utilizing XPS to characterize the chemical composition of nano drugs and employing Western blotting (WB) techniques to confirm successful cell membrane encapsulation.

3.      The cellular uptake needs to be quantitatively processed.

4.      This study only demonstrated that the design of nano drugs enhances cellular uptake. However, whether this could adversely affect normal cells remains unclear. Therefore, the biological safety of nano drugs needs to be assessed. Additionally, can loading anti-inflammatory drugs enhance the anti-inflammatory performance of these nano drugs?

Comments on the Quality of English Language

minor

Author Response

Q1: Cell membrane modification technology is the focus of drug design in this article. Therefore, it is recommended that the authors provide a comprehensive overview of cell membrane technologies in the Introduction section. This should include the current treatment status of cancer (Science Advances, 2024, 10, eadm9561; Small, 2024, 2404741), inflammation (Nature Communications, 2024, 15, 1042; Advanced Science, 2024, 2310211.), and other diseases (Biomaterials, 2024, 306, 122478; Exploration, 2024, 20230164; Advanced Science, 2023, 10, 2302131) using cell membrane-wrapped nano drugs administered via intravenous injection, oral ingestion, and other delivery routes. Additionally, they should summarize the advancement of current research methods in this field.

A1: We thank the Reviewer for the comment. We have revised the introduction to include an overview of cell membrane technologies, incorporating the recommended articles.

Q2: The morphology and composition characterization of nano drugs need further refinement. For example, utilizing XPS to characterize the chemical composition of nano drugs and employing Western blotting (WB) techniques to confirm successful cell membrane encapsulation.

A2: We would like to extend our gratitude to the Reviewer for the comment. Regrettably, it is not possible to conduct XPS analysis on our samples due to their composition, which is based on a liquid core of secondary nanoemulsions. XPS analysis requires dry samples, and our NEsoSome starts to degrade if they are dried. Additionally, the chemical composition which can be assed with XPS analysis closely resemble that of biological membranes, containing elements such as carbon (C), oxygen (O), phosphorus (P), and nitrogen (N). The same considerations can be made for WB analysis, in which the pre-treatment of our NEsoSome could destroy the architecture. The perfect coating of our nanocarrier is supported by Cryo-TEM analysis, clearly demonstrating membrane deposition. Additionally, the increase in size value and the variation of zeta potential values confirm the membrane deposition.

Q3: The cellular uptake needs to be quantitatively processed.

A3: We thank the Reviewer for the insightful feedback. We have now reported the plot of the mean fluorescence intensity (MFI) normalised for the cell number. After 24 hours, the MFI reaches a plateau, indicating maximum internalisation for both M0 NEsoSome and M-NEsoSome. The uptake efficiency of particles in HUVEC cells and TNF-α-treated HUVEC cells improved for both M0 NEsoSome and M-NEsoSome, supporting the hypothesis that all formulations were fully internalised after 48 hours. The highest intensity was observed for M0 NEsoSome in TNF-α-treated HUVEC cells.  In healthy HUVECs, both biomimetic formulations (M0 NEsoSome and M-NEsoSome) behaved similarly during the initial 30 minutes and up to 24 hours. However, after 24 hours, a slight difference was observed, with M-NEsoSome showing better internalisation.

Q4:This study only demonstrated that the design of nano drugs enhances cellular uptake. However, whether this could adversely affect normal cells remains unclear. Therefore, the biological safety of nano drugs needs to be assessed.  Additionally, can loading anti-inflammatory drugs enhance the anti-inflammatory performance of these nano drugs?

A4: We thank the Reviewer for the insightful feedback. We appreciate his/her interest in the biological safety and the potential enhancement of the anti-inflammatory performance of our nano drugs. The current study primarily focused on demonstrating the enhanced cellular uptake of the nano drugs; we recognise the importance of assessing their potential adverse effects on normal cells. In previous studies, we have worked with similar types of nanocarriers, employing various cellular membranes. Previous and present studies included rigorous uptake analyses, which did not indicate signs of cytotoxicity or mortality in normal cells (Pharmaceutics 202113, 1069). Moving forward, we plan to conduct comprehensive biocompatibility and toxicity assessments to thoroughly evaluate the safety profile of our nano drugs in both normal and pathological conditions. Regarding your query about loading anti-inflammatory drugs, as also shown in previous studies (ACS Nano 2017, 11, 10, 9802–9813), the incorporation of anti-inflammatory agents into our nanocarriers could potentially amplify their therapeutic efficacy. We believe that these additional studies will address the safety concerns and provide a more comprehensive understanding of the therapeutic potential of our nanocarriers. We have now added this consideration in the revised version of the manuscript.

Figure 1 Plot of mean fluorescence intensity of nanocarrier normalised to cell number. TNFα HUVECs (A) and HUVECs (B) were treated with Ct NEs, M-NEsoSome and M0-NesoSome at different time points: 30min, 24h and 48h. Data are reported as mean (n=5) ± SD.

Reviewer 2 Report

Comments and Suggestions for Authors

E. Lagreca et al. developed two biomimetic nanocarriers (NCs). The NC used oil in water
Nanoemulsion (SNE) as the inner core for high lipophilic compound loading. NC uptake ability was investigated in tumor necrosis factor alfa (TNFα)- treated human umbilical vein endothelial cells (HUVECs). Please address the below comments.

1.      The introduction section lacks a problem analysis. A clear research gap should be described. The last paragraph should convey the novelty and gist of the research, rather than an overview of the study.

2.      There are a few typographical errors throughout the manuscript, for example, section 1.2.1. the term CO2, 5x10^5 cells/mL, 10%, 37°C, etc. Please correct such mistakes.

3.      It is important to evaluate the cytotoxic effects of the NCs, please perform the cell viability assessment.

4.      The authors did not discuss the stability of NCs.

Comments on the Quality of English Language

There are a few typographic errors. 

Author Response

Q1: The introduction section lacks a problem analysis. A clear research gap should be described. The last paragraph should convey the novelty and gist of the research, rather than an overview of the study.

A1: We thank the Reviewer for the constructive comment. We have revised the introduction to include an overview of cell membrane technologies.

Q2: There are a few typographical errors throughout the manuscript, for example, section 1.2.1. the term CO2, 5x10^5 cells/mL, 10%, 37°C, etc. Please correct such mistakes.

A2: We thank the Reviewer for his/her comment; we corrected the typographical errors throughout the manuscript.

Q3: It is important to evaluate the cytotoxic effects of the NCs, please perform the cell viability assessment.

A3: We thank the Reviewer for the constructive feedback. We appreciate his/her emphasis on the importance of evaluating the cytotoxic effects of the nanocarriers (NCs). We fully agree that assessing the cytotoxic effects of NCs is crucial to ensuring their safety and biocompatibility. While our current study primarily focused on demonstrating enhanced cellular uptake, we recognise that a thorough evaluation of cell viability is essential to validate the safety profile of our NCs. In our previous research, we have utilised similar types of nanocarriers with various cellular membranes. Previous and present studies included detailed uptake analyses, and we did not observe cytotoxicity or mortality in normal cells. This provides a preliminary indication of the biocompatibility of our NCs.  Moving forward, we plan to integrate these cytotoxicity studies into our research framework. This will not only validate the safety of our NCs but also provide critical insights into their therapeutic potential and dosage optimisation. We are committed to ensuring the highest standards of safety and efficacy in our research and will prioritise these evaluations in our ongoing and future studies.

Q4: The authors did not discuss the stability of NCs.

A4: In response to the concern about the stability of our system, we evaluated the course of one month by storing the samples at 4°C, as illustrated in the figure. After 4 weeks, we observed an increase in size for the layered NEsoSome of up to 80 nm, indicating the beginning of the destabilisation process.

Figure 1 Dimensional stability over the time of Ct NEs, M-NEsoSome and M0-NesoSome. Data are reported as mean (n=3) ± SD.

Reviewer 3 Report

Comments and Suggestions for Authors

    Dr. Lagreca et al investigated the application of monocyte and macrophage membrane for the biomimetic nano emulsion facilitating to target into the endothelial cells. This novel strategy that monocytes and macrophage membrane is use for drug targeting materials is interesting.

   At the present stage, the findings is preliminary results using the in vitro cell line, so it is highly desirable to obtain advantageous findings from in vivo experiments including human as the future study. Reviewer has raised the following concerns and would appreciate it if authors could address them.

In Figure 5, the intensity for M0-NEsoSome was only low value in comparison to that for Ct-NEs and NEsoSome. Why is there such a difference?

In Figure 7 and 8, The imaging results looks represented the enhancement of uptake of particles in the each HUVEC cells in the cases of M and M0 MEsoSome. Since it is difficult to recognize the extent to which the uptake efficiency has been improved only from the image results, please provide a quantitative data of how much the uptake efficiency has improved.

Author Response

Q1:   At the present stage, the findings is preliminary results using the in vitro cell line, so it is highly desirable to obtain advantageous findings from in vivo experiments including human as the future study. Reviewer has raised the following concerns and would appreciate it if authors could address them.

 A1: We thank the Reviewer for his/her valuable feedback. We acknowledge that our current study presents preliminary results based on in vitro cell line experiments. These initial findings have demonstrated the potential of our nanocarriers (NCs) in enhancing cellular uptake and providing therapeutic benefits in a controlled laboratory setting. We agree on the necessity of validating our findings through in vivo studies. Our future research will include extensive in vivo experiments to assess the biodistribution, pharmacokinetics, efficacy, and safety of our nanocarriers (NCs). We are committed to addressing these concerns in our ongoing and future studies to ensure the safe and effective development of our NCs.

Q2: In Figure 5, the intensity for M0-NEsoSome was only low in comparison to that for Ct-NEs and NEsoSome. Why is there such a difference?

A2: We thank the Reviewer for his/her valuable feedback. We improved the resolution of images in order to better understand the difference in intensity among the different samples. The different values of intensity are reported in the subsequent response.

Q3: In Figure 7 and 8, The imaging results looks represented the enhancement of uptake of particles in the each HUVEC cells in the cases of M and M0 MEsoSome. Since it is difficult to recognize the extent to which the uptake efficiency has been improved only from the image results, please provide a quantitative data of how much the uptake efficiency has improved.

A3: We thank the Reviewer for his/her observation. We have now reported the plot of the mean fluorescence intensity (MFI) normalised for the cell number. After 24 hours, the MFI reached a plateau, indicating maximum internalisation for both M0 and M-NEsoSome. The uptake efficiency of particles in HUVEC cells and TNF-α-treated HUVEC cells improved for both M0 and M-NEsoSome. This supports the hypothesis that all formulations were fully internalised after 48 hours, with the highest intensity observed for M0-NEsoSome in TNF-α-treated HUVECs (Figure A).

In healthy HUVECs (Figure B), the biomimetic formulations behaved similarly for the first 30 minutes and 24 hours. However, a slight difference was observed after 24 hours, with M-NEsoSome showing better internalisation.

Plot of mean fluorescence intensity of nanocarrier normalized to cell number. TNFα HUVECs (A) and HUVECs (B) were treated with Ct NEs, M-NEsoSome and M0-NesoSome at different time points: 30min, 24h and 48h. Data are reported as mean (n=5) ± SD.

Round 2

Reviewer 1 Report

Comments and Suggestions for Authors Most comments given by the reviewers have been addressed. This manuscript could be accepted now.